# Dispatching Ambulances using Deep Reinforcement Learning

## Abstract

Emergency Medical Service (EMS) plays an essential role in today's society. One EMS component is ambulance dispatch, which impacts the ambulance's response time for a medical incident. Fast response times are essential. The problem of ambulance dispatching differs from a typical Vehicle Routing Problem (VRP) since patients arrive stochastically, making the problem hard to solve. In addition to minimizing response time, EMS providers seek optimal resource utilization and good working conditions for EMS personnel while often experiencing an increase in demand. To meet these requirements, this work develops a Reinforcement learning (RL) method based on Proximal Policy Optimization (PPO) for the ambulance dispatching problem. Varying incident priorities along with more flexible incident queue management are also integrated into our novel method. Our PPO-based method and an EMS simulation model are implemented in Python and combined with Open Street Map (OSM) travel time estimation and simple synthetic incident data generation. Empirical results are presented using both synthetic and real incident data. Results using real incident data from the Oslo University Hospital (OUH) in Norway suggest that our PPO model outperforms heuristic policies such as dispatching the closest ambulance by Haversine or Euclidean distance. We hope that this work inspires future research on RL for ambulance dispatch and ultimately leads to improved decision-support tools for EMS in Norway and elsewhere.

## 1 Introduction

**Context.** The Emergency Medical Service (EMS) at the Oslo University Hospital (OUH) in Norway is handled by its Emergency Medical Communication Centre (EMCC). Fast response times are essential, especially for acute incidents including cardiac and circulatory arrest. In cardiac and circulatory arrest incidents, chances for resuscitation drop fast over time. This change has an estimated drop of 10% every minute. In addition, OUH has experienced an increase in incidents in recent years, perhaps due to a steady population increase over time. At the same time, EMCC resources are limited, and working conditions can be stressful. At times, EMS and ambulance personnel may need to work for long hours without breaks or socializing. Therefore, one also needs to carefully consider the human and social aspects of EMS.

**Description of Problem and Challenges.** We now consider how an EMS system works, including the problem of ambulance dispatch. Ambulances are at their base stations when not on assignment, and their distribution among base stations is called allocation. Incidents are by the EMCC triaged into three different priority levels: acute ($A$), urgent ($H$), and regular ($V$). The priority reflects how fast an ambulance should respond to an incident (the response time). Regular incidents, the lowest-priority incidents, are not considered in this work. A high-level perspective of the steps taken for an incident is as follows:

1. An incident occurs, and someone calls the EMCC.

2. A dispatcher at the EMCC assesses the priority $i_p \in \{A, H\}$ of the incident.

3. An ambulance close to the incident is dispatched to the incident's location. The haste is defined by the given priority. Exactly which ambulance to dispatch is an important but difficult decision, which has been researched previously and is what we study in this paper.

4. The ambulance arrives at the incident location and either treats the patient on the spot or picks up the patient. The ambulance response time, which clearly depends on which ambulance is dispatched to this incident, is recorded as $t_r$.

    4.1 If not treated on the spot, the ambulance delivers the patient to a hospital.

5. After treating the patient, the ambulance drives to its assigned base station.

The Norwegian Directorate of Health has set goals on how fast the response time should be, and indirectly this places demands on ambulance dispatch as well as other EMS operational factors. These goals are quantifiable and depend on the incident's priority and the population density. The goals revolve around the CDF of the response time: (i) In densely populated areas 90% of acute incidents should have a response time lower than 12 minutes. (ii) In sparsely populated areas 90% of acute incidents should have a response time lower than 25 minutes Schjølberg & Bekkevold (2022) Bib (2023). Additionally, there is the question of how to optimize the computer-aided dispatch (CAD) software used for EMCC decision support. Overall, the challenge of providing high-quality EMS, including ambulance dispatch, is to meet ambitious goals on response time while carefully considering the human and social aspects of EMS workplaces as well as resource restrictions (including a restricted number of ambulances and base stations).

**Contributions.** We now consider our main contributions compared to previous research. Primarily the novelty in this work lies in the integration of the many complex real world aspects of the ambulance dispatching problem. Especially the integration of incident priority, incident queue, ambulance shifts and ambulance availability in unison. Integration of these aspects in a RL aspect is hard, due to the many dimensions such as time, location, availablity and priority.

- A main contribution is our adaptation of Proximal Policy Optimization (PPO) to the ambulance dispatch setting. While Bélanger et al. (2019) highlights a need for experimentation with new RL methods for emergency medicine, existing research uses more traditional methods including approximate dynamic programming (ADP) Schmid (2012) Nasrollahzadeh et al. (2018), Markov decision process (MDP) McLay & Mayorga (2013) Hua & Zaman (2022), semi-Markov decision process (SMDP) Mukhopadhyay et al. (2019), or temporal difference learning (TDL) Hua & Zaman (2022). PPO has not been applied to the problem of ambulance dispatching before, although Holler et al. (2019) apply PPO to a similar problem and achieve promising results.

- Some of the literature considers both ambulance relocation and dispatching problems Nasrollahzadeh et al. (2018)Schmid (2012). However, there is little consideration of changing the number of available ambulances during day and night shifts, as is done here.

- Incident priority is often ignored in previous research, to reduce the complexity of the ambulance dispatching problem Liu et al. (2020) Mukhopadhyay et al. (2019) Elfahim et al. (2022)Schmid (2012). Nasrollahzadeh et al. (2018) and Bandara et al. (2012) were the only found papers found which considers RL and incident priority to ambulance dispatching. In contrast, we integrate incident priority since it affects the dispatching order of the ambulances. We integrate a make shift survival function as reward which considers both incident priority and their relation to reponse time.

- In brief, previous work uses a FIFO incident queue McLay & Mayorga (2013)Schmid (2012)Mukhopadhyay et al. (2019) while we can dispatch any queued incident. Some solutions use a priority-sorted incident queue Nasrollahzadeh et al. (2018).

- Due to potential overfitting when using only historical data, we developed a synthetic incident generator. We test our model on both historical and synthetic data reflecting incidents reported to OUH's EMCC, to ensure proper validation.

- Open Street Map is utilized for simulation which is more reproducible than previous work.

## 2 BACKGROUND

**Reinforcement Learning (RL)** is a paradigm in machine learning that differs from unsupervised and supervised learning. It models intelligent agents interacting with an environment to maximize their reward Sutton & Barto (2018). In this environment, states, possible actions, and rewards are

outlined. Given a state of the environment $s$ it transitions into a state $s'$ after performing action $a$ and receives reward $r$. The goal is to perform actions through time $t$ to maximize an episode's discounted reward. An episode is a series of states and actions in an environment instance. An episode can terminate when a terminal (goal) state is achieved, or a maximum number of iterations limit is reached. When the environment is non-deterministic, the state transitions from $s$ to $s'$ with action $a$ with a probability $P(s \rightarrow s'|a)$, often denoted $P(s'|s, a)$.

An action $a$ is modeled through a policy $\pi$ learned through time. Policies utilize a state-value function $V_\pi(s)$, which models the desirability for an intelligent agent to be in state $s$ (following policy $\pi$). The state-value function is the expected future rewards when actions are performed from that state. These future rewards are discounted using a coefficient $\gamma$. With an infinite time horizon, the state-value function is:

$$V_\pi(s_{t=0}) = \mathbb{E}[\sum_{t=0}^{t=\infty} \gamma^t \cdot r_t] \tag{1}$$

**Actor Critic Model** is a policy gradient method composed of two models, extending the Temporal Differencing (TD) and RL theory. The actor embodies the policy denoted as $\pi$, whereas the critic embodies the state value function $V(s)$. The critic evaluates how well the actor performs in the environment through TD-error $\delta$ (Also denoted $A$ for advantage in literature). These two models are trained concurrently, so the critic becomes better at evaluating the actor while the actor's policy improves over time. Sutton & Barto (2018).

**Proximal Policy Optimization (PPO)** is an on-policy, online policy gradient method Schulman et al. (2017). It is similar to the advantage actor-critic model but ensures stability by restricting its value updates to not deviate too much from the old policy. It does this by clipping the probability ratio between the new and old policy.

$$b(\theta) = \frac{\pi_\theta(a_t, s_t)}{\pi_{\theta_k}(a_t, s_t)}. \tag{2}$$

While the PPO model trains the actor, a hyperparameter $\epsilon$ defines a trust region. The trust region $1 + \epsilon$ and $1 - \epsilon$ defines how much the new policy $\pi_\theta$ can deviate from the previous policy $\pi_{\theta_k}$. The ratio of change in probability is defined by Equation 2, high $b(\theta)$ means that action $a_t$ is much more likely in $\pi_\theta$ than in $\pi_{\theta_k}$. The clipping of this ratio is performed by $c(\theta, \epsilon) = clip(b(\theta), 1 - \epsilon, 1 + \epsilon)$, which ensures $b(\theta)$ comprises of values between $1 - \epsilon$ and $1 + \epsilon$.

The loss function of the PPO algorithm (Equation 3) is similar to the loss function of the advantage actor-critic model but ensures stable policy updates. The main logic is that when the performed action $a$ was reasonable ($\delta_t > 0$), the policy update is limited to the trust region. The incentive to move $b(\theta)$ beyond the clipping range is removed.

$$J^{clip}(\theta) = \sum_{t=0}^{t=T} min(b(\theta)\delta_t, c(\theta, \epsilon)\delta_t) \tag{3}$$

**Ambulance Dispatch.** Computational problems that arise in connection with EMS include ambulance allocation McCormack & Coates (2015) Schjølberg & Bekkevold (2022), demand forecasting Setzler et al. (2009) Zhou (2015) Hermansen (2021) Van De Weijer & Owren (2022), and ambulance dispatch Bélanger et al. (2019) Neira et al. (2022) and Mukhopadhyay et al. (2020). We study ambulance dispatch in this work. Various optimization methods have been applied to the dispatching problem besides RL. These other methods include linear programming McLay & Mayorga (2012), mixed integer programming Albert (2022), tabu-search Li & Saydam (2016), and genetic programming MacLachlan et al. (2023). Our focus is on RL-based dispatch, with previous research on RL for ambulance dispatching summarized in Table 1. Somewhat similar RL approaches exist for vehicle fleet management (typically taxi fleet management), ride-sharing, and ride-hailing, but given space limitations we do not discuss them here. While RL training can be slow, there are several potential benefits for ambulance dispatch, which can be summarized as follows: RL can capture the stochastic and dynamic dimensions of EMS; RL works well even when spatial prediction is hard Qin et al. (2021); RL can work online (depending on the algorithm) and adapt to changing circumstances; once trained RL inference time is fast; and the model can be updated after deployment. From an RL perspective, the ambulance dispatch problem is still complex: The action and state spaces are

typically large, the environment is stochastic and dynamic, and multiple agents (ambulances) are involved.

| Citation | Year | Method(s) | Focus | Reward | Environment | Training method | Online |
|---|---|---|---|---|---|---|---|
| Schmid (2012) | 2012 | ADP | Pure dispatching | Response time | CAD data / Synthetic | Value iteration | No |
| McLay & Mayorga (2013) | 2013 | MDP | Patient classification errors | Coverage | CAD data | Uniformization | No |
| Nasrollahzadeh et al. (2018) | 2018 | ADP | Dispatching / relocation | Priority adjusted response time | CAD data / Synthetic | Approximate policy iteration | No |
| Mukhopadhyay et al. (2019) | 2019 | SMDP | Complete pipeline | Response time | CAD data / Synthetic | MCTS | Yes |
| Liu et al. (2020) | 2020 | DQN, MAQR | Pure dispatching | Wait time Count incidents Round trip time | CAD data / Synthetic | Experience replay | No |
| Elfahim et al. (2022) | 2022 | DQN | Pure dispatching | Response time | CAD data | Experience replay | No |
| Hua & Zaman (2022) | 2022 | MDP, TDL | Pure dispatching Augmented transition probabilities | Response time | N/A | Policy iteration | No |

Table 1: Previous research on ambulance dispatching using RL. Methods used include approximate dynamic programming (ADP), deep Q-Network (DQN), Multi-Agent Q-Network with Experience Replay (MAQR), Markov decision process (MDP), semi-Markov Decision Process (SMDP), and temporal difference learning (TDL). The rightmost column refers to whether online policy updates are done or not.

**Choice of model.** In older literature, it is more common to use mdp models; a more common trend is to use more sophisticated model-free models such as ADP, MAQR and DQN. All of these models face the issue of the ambulance dispatching problem being a non-homogeneous Markov decision process. In other words, the state transition probabilities have temporal instability.

The Markov Decision Process of the ambulance dispatching problem can also be seen as partially observable (POMDP). Since the state, which includes the current incident locations, might not unveil enough information about the future incident distribution.

An assumption for the usage of Markov Decision Process in general is that the Markov property holds for the environment, which might not be the case for the ambulance dispatching problem. Since the future spatial distribution of incidents might depend on several past states rather than only the current state (Markov property). This is why papers like Mukhopadhyay et al. (2019) use techniques like Monte Carlo Tree Search (MCTS), which relaxes the Markov property.

Furthermore, Mukhopadhyay et al. (2019) uses SMDP, which fits better to the problem of ambulance dispatching than MDP. Since in SMDP the time between decision-making states (ambulance needs to be dispatched) can be random Cochran et al. (2010). In MDP, this time (sojourn time or time-step) is assumed to be static for each state.

Considering all of this, policy gradient methods Morimura et al. (2022) (such as PPO), other model-free RL methods (such as MAQR and Deep Q-network (DQN)), and MCTS might suit the problem more. These methods relax the Markov property and approximate the state transition probabilities.

## 3 DATASETS

**Datasets.** Two big datasets are used in this work. The first is a dataset with ambulance incidents retrieved from OUS, while the second dataset contains the road network of Oslo and Akershus, retrieved from Open Street Map (OSM). These datasets are discussed further below.

**Incident dataset.** The Incident dataset is an anonymized dataset of ambulance incidents, meaning the location of the incidents is aggregated into 1x1 km grids of Oslo and Akershus. The dataset contains 2597 such grids with about 752k incidents ranging over eight years from 2001 to 2019 (2001, 2002, 2005, 2015-2019). The grids in the dataset are combined with population data from Statistics Norway (SSB). This dataset contains extensive meta-information about the incidents and ambulances dispatched. For example, it includes which type of ambulance was sent, ambulance id, grid location of the incident, and incident priority. It also contains timestamp information about the different stages of the EMS process. However, the location of the dispatched ambulance is missing.

**Open Street Map (OSM)** is an open-source map tool which provides their maps available for download. It is freely available for all interested parties, and the maps it provides are also updated regularly. The maps can also be downloaded as a PBF file, which then can be analyzed for research

purposes. OSM provides one such map file for Oslo and one for Akershus. These two files were merged to create a complete map of Oslo and Akershus. The roadnetwork is represented as a directed graph.This dataset is used for ambulance travel time estimation. A considerable amount of preprocessing was performed to make the graph tractable, and to align with the granularity of the incident dataset. The travel time between two cells of the incident dataset is calculated based on random sampling from each cell. The resulting graph is referred to as Grid-cell graph H.

## 4    DATA ANALYSIS

In this section, some data analysis of the real-world incident dataset is shown. Further data analysis can be seen in Schjølberg & Bekkevold (2022), Van De Weijer & Owren (2022), Hermansen (2021) and Hermansen & Mengshoel (2021) which have utilized the same dataset.

**Response time.** The response time for a given incident $i$ varies greatly depending on which priority $i_p \in \{A, H\}$ is assigned. As seen on Figure 1 (left), Acute incidents have a much lower response time on average than Urgent. Formally $(\overline{t_r}|i_p = A) \approx 12$ and $(\overline{t_r}|i_p = H) \approx 25$ minutes, which implies $(\overline{t_r}|i_p = A) < (\overline{t_r}|i_p = H)$.

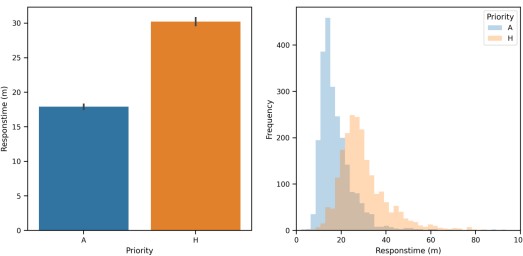

Figure 1: (Left) Average response time per priority, with marked 95% confidence interval. (Right) histogram plot of response times per incident.

**Estimated survival function.**    There is a non-linear relationship between response time and chance of patient survival, which is described by a survival function. Such a survival function can be used to describe the relative importance of an incident (survivability) given response time $t_r$ and incident priority $i_p$. In other words, these describe how much more important each incident priority is compared to each other as a function of $t_r$. Unfortunately, OUH does not have such a survival function for Oslo and Akershus, therefore a survival function for each priority is estimated. The constructed survival functions is the inverse CDF of the response time $t_r$ for each priority $i_p$, estimated from the dataset (Figure 2).

## 5    METHODS AND MODELS

Figure 3 shows an overview of the main components. These components include the RL model and Simulator which interact with each other through states, rewards, and actions. The RL model is composed of both Actor and Critic networks. The Simulator uses the preprocessed grid-cell road network graph $H$, incident data, and the synthetic incident data generator. Lastly, the input and output of the RL model and Simulator are highlighted.

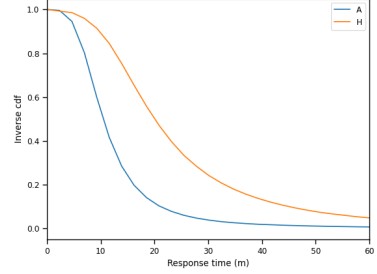

Figure 2: The inverse CDF of the response time for the two priorities Acute (A) and Urgent (H). In other words, the estimated survival functions for the two priorities.

### 5.1    SIMULATION

A simulator model was built based on knowledge gained from the incident dataset and data preparation of the OSM dataset. This simulator model was made to reflects a general EMS dispatch setup

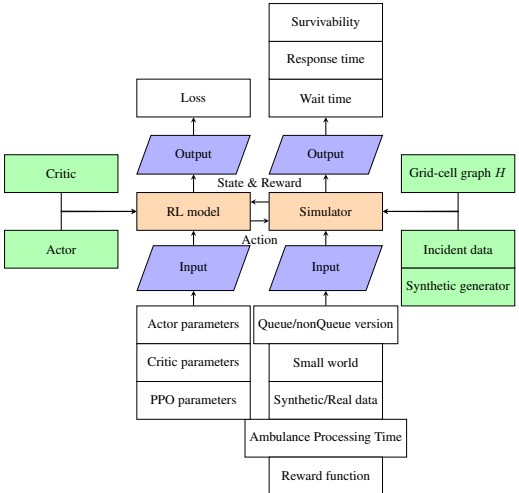

Figure 3: Shows the inputs and outputs of the different parts of the framework utilized. Elements in green are the components that RL and simulator consists of.

and counteracts the fact that ambulance locations are lacking from the incident dataset. Furthermore aspects such as incident queue, time spent by ambulances not driving (handling equipment etc. here called ambulance processing time) and ambulance day/night shifts are simulated.

### 5.1.1 SYNTHETHIC AMBULANCE DATA

The synthetic generator assumes that incidents arrive with a global constant $\lambda = 0.21$ in the exponential probability distribution; $P(X = x) = \lambda e^{-\lambda x}$. This is then used as a discrete PMF to draw $\delta t = X$, which is used to derive the location of the incident[1] $i_l$.

To get the location of the incident (*genIncident* (Algorithm 1)), a homogeneous Poisson process is assumed for all the locations. Each of these processes follows the exponential distribution, and the parameters are collected into a vector $\lambda_L = (\lambda_{l=0}, \dots, \lambda_{l=n})$. In other words, one element from $\lambda_L$ represents the mean $\lambda$ for a single location. These parameters are used to derive the location of the incident once an incident has occurred (determined by the global $\lambda = 0.21$).

---

**Algorithm 1** *genIncident*, generate synthetic incident

**Require:** $\delta t$ time since the last incident occurred, $\lambda_L = (\lambda_{l=0}, \dots, \lambda_{l=n})$ lambda for each grid-cell location, $T = (t_{l=0}, \dots, t_{l=n})$ times since incident happened at each grid-cell location (all zero).
**Ensure:** Generated location and priority of the incident.
  1: $T = T + \delta t$                    ▷ Update times since last incident occurred
  2: $P = 1 - e^{\lambda_L T}$                    ▷ Update probabilities
  3: $P = \frac{P}{\sum P}$                    ▷ Normalize
  4: $i_l = \text{Random}(P, 1)$          ▷ Draw one random location, from discrete PMF
  5: $T_{l=i_l} = 0$          ▷ Set time since last incident occurred at this location to zero
  6: $i_p = \text{Random}((A = 0.57, H = 0.43), 1)$                    ▷ Draw incident priority
  7: **return** $T, i_l, i_p$

---

### 5.1.2 REAL-WORLD AMBULANCE DATA

The incident dataset $\mathcal{D}$ has records $\mathcal{D} = (r_0, r_{\dots}, r_n)$. Where each record contains $(i_t, i_l, i_p, \text{t-d}, \text{i-d}, \text{h-l}, t_n)$ tuples. This record denotes that an incident $i$ occurs with priority $i_p$ at grid-cell $i_l$ at time $i_t$. The dataset is sorted by increasing $i_t$. Furthermore, t-d, i-d, and h-l denote

---

[1]We use the same notation as for the incident dataset, but incidents here are synthetically generated

the different processing times for this incident. t-d is the amount of time between an ambulance has been assigned until it starts driving, i-d is the amount of time between an ambulance arrives at the incident location until it starts driving, h-l is the time between the ambulance arrives at the hospital until it starts driving again. Lastly, $t_n$ is the time until the next incident occurs, which the RL model does not know.

The records from the dataset $\mathcal{D}$ are split into a time series train-evaluation split $\mathcal{D}_t, \mathcal{D}_e$. This is performed such that $\frac{|\mathcal{D}_t|}{3} \approx |\mathcal{D}_e|$, and $(\forall_{r_t \in \mathcal{D}_t} i_t \in r_t) < (\forall_{r_e \in \mathcal{D}_e} i_t \in r_e)$. This essentially means that the size of the evaluation set is one third of the training set, and comes after the training set in time $i_t$. Which type of data used for training and evaluation is outlined in each experiment in the next section.

## 5.2 REINFORCEMENT LEARNING (RL)

### 5.2.1 STATE

The state provided by the simulator is based on the location of the ambulances and incidents. Since the RL model supports multiple state inputs, the model provides one list of the ambulance locations $A$ and one list for the incident locations $I$.

The size of these lists equals the number of locations (grid-cells) $N$ simulated in the model ($|A| = |I| = N$). The ambulance list $A$ is defined by $A = (f_{ak})_{k \leq N}$, where the value of $f_{ak}$ is the number of available ambulances located at location $k$. Similarly, The incident list $I$ is defined by $I = (f_{ik})_{k \leq N}$, where the value of $f_{ik}$ is the number of incidents located at location $k$. This makes $\sum I$ the current number of incidents and $\sum A$ the number of available ambulances. When multiple ambulances are located at a base station location $k$, then $f_{ak} > 1$. Further, there can be more than one incident in the incident list if the queue model is used ($\sum I > 1$).

### 5.2.2 ACTION

In the non-queue model, an action is a choice between available ambulances. In other words, the action space equals the number of simulated ambulances, but only available ambulances can be chosen by the RL model in any given state.

In the queue model, the action space equals the Cartesian product $A \times I$. In other words, both an available ambulance and an incident are chosen by the RL model.

### 5.2.3 REWARD

In a given state $s$, the reward $R(s)$ is defined as the negative sum of the waiting times $w_t(i)$ (Equation 4). The waiting time for an incident increases until an ambulance arrieves.

$$R(s) = -\sum_i w_t(i).$$
(4)

This reward function does not consider incident priority. Drawing inspiration from Bandara et al. (2012), this is achieved by combining the estmated survival function and Equation 4 into Equation 5.

$$R(s) = \prod_i f(w_t(i), i_p).$$
(5)

Here, $i_p$ is the priority of incident $i$, and $f$ is the estimated survival function. Rather than using response time threshold, this method estimates more directly the relative importance of each priority as a function of response time.

## 6 EXPERIMENTAL RESULTS

Mainly two experiments were performed. The first experiment trains on synthetic data, and evaluates on the testset $\mathcal{D}_e$. The second experiment trains on the train set and evaulates on the test set $\mathcal{D}_e$. A small portion of Oslo and Akershus is considered for simulation, taking a 5 km radius from Oslo central station.

The trained RL model is compared to three other dispatching agents. The first agent dispatches the ambulance with the lowest Haversine distance (Haversine), while the second agent according to the lowest Euclidean distance (Euclidean). Finally, the last agent dispatches a random available ambulance (Random). These policies are shown in the results in the next section, differentiated by the name in parentheses.

It is a possibility that the RL model dispatches the same ambulance as the Haversine policy. Hence the fraction of such actions performed is kept track of (Haversine fraction). This shows how different the trained policy is from the Haversine policy.

## 6.1 Tools

The Python packages Stable Baselines (Raffin et al. (2021)) and GeoPandas (Jordahl et al. (2020)) is utilised throughout this work. Stable Baselines is used for RL, while GeoPandas is used for preprocessing of the OSM road network dataset.

## 6.2 Experiment 1: From Synthetic to Natural Data

In experiment 1 the model is trained on synthetic incidents and evaluated on $\mathcal{D}_e$. The goal is to study whether the model is able to adapt to the real test data when trained on synthetic data. Incident queue, Incident priority and ambulance processing time (time not spent on driving) are not considered in this experiment. Lastly the standard wait-time reward is used, see Equation 4. The results are shown in Table 2

| Model | MRT (m) | MR | HF | Iter. |
|---|---|---|---|---|
| RL | 16.78 | -0.67 | 0.23 | 29052 |
| Haversine | 17.29 | -1.69 | 1.00 | 29052 |
| Euclidean | 17.36 | -1.69 | 0.84 | 29052 |
| Random | 18.18 | -2.05 | 0.22 | 29052 |

Table 2: *Results from **Experiment 1** for four different models with varying Haversine fraction (HF), using the same number of iterations. The mean reward (MR) should be as high as possible, while the mean response time (MRT) should be as low as possible.*

Overall the RL agent outperforms the other agents in this experiment. The difference between Euclidean and Haversine is slim, but the Haversine performs best.

## 6.3 Experiment 2: Incident Prioirities

In experiment 2 the model is trained on $\mathcal{D}_t$ and evaluated on $\mathcal{D}_e$. Incident priority is considered, which implies that the estimated survival function is utilized. Furthermore, ambulance processing time (which increases response time), ambulance shifts, and incident queue are also simulated.

This setup also makes use the Cartesian product between the incidents and available ambulances as action space. This makes it possible for the RL model to choose any incident in the queue or not. The state space also includes a survivability list, which contains the estimated survival probabilities for each incident (subsubsection 5.2.1).

| Model | MRT (m) | MR | HF | Iter. |
|---|---|---|---|---|
| RL | 23.31 | 0.57 | 0.19 | 29052 |
| Haversine | 28.88 | 0.42 | 1.00 | 29052 |
| Euclidean | 29.13 | 0.41 | 0.74 | 29052 |
| Random | 26.42 | 0.46 | 0.23 | 29052 |

Table 3: *Shows results from **Experiment 2**. The Mean reward (survivability) should be as high as possible, while the Mean response time should be as low as possible.*

Based on the results shown in Table 3 and Figure 4, the response time of the random agent is lower than for both haversine and euclidean distance. This experiment had several runs with the same result. This is most likely due to the small environment considered. The RL agent out-performs the other evaluated policies. Judging from the histogram, the Haversine agent has slightly more lower reponse times ($< 25m$) than the Random and Euclidean agent. The high **Mean response time** and low **Reward** for these agents is probably because they have no consideration for the priority or for the incident queue.

The RL model seems to have not overfitted on $\mathcal{D}_t$. There is, however, significant doubt that this RL model considers the future incident distribution. Since the state space provided provides little information about the history of the incident distribution. If this was the case, the synthetic data generator should be used in combination with $\mathcal{D}_t$ to reduce overfitting RL model assumed future incident distribution.

## 7    CONCLUSION AND FUTURE WORK

This work has explored the usage of RL and PPO to the ambulance dispatching problem as a potential decision-support tool for the EMS. The main contribution is that we include incident priority, incident queue, day/night shifts, usage of PPO, extensive data analysis, and reproducible OSM travel time estimation.    Finally, a simple method for synthetic incident data generation is also provided.    A literature overview is outlined with a table of the most recent literature on the usage of RL to the ambulance dispatching problem.    Finally, a discussion of these implementations and recent trends is outlined.

This work has also shown how OSM can be preprocessed to be used for scientific purposes. Extensive work went into the preprocessing to make the data tractable. Furthermore, extensive work went into building a simulator for the ambulance dispatching problem. This simulator considers many aspects of the ambulance dispatching problem, such as incident priority, incident queue, ambulance shifts, small world setup, Synthetic/Real incident data, and ambulance processing time. This simulator uses a time step equal to the incident interval time, which can speed up the simulation. Furthermore, it is implemented in Python, making it more suitable for Machine Learning.

Overall, our PPO model consistently outperforms the other heuristic agents evaluated throughout the experiments. Generally, the Haversine distance mostly outperformed the Euclidean distance when considering Mean response time; however, this difference is extremely slim (9.8s saved on avg). This shows that RL and road network travel time have great potential to reduce response time and increase survivability. However, more work is needed for RL to be implemented as a decision-support tool for the EMS. While our RL model did not overfit the training dataset, a more complex model might do so. Therefore, combining synthetic and real data is recommended during training is recommended for future studies.

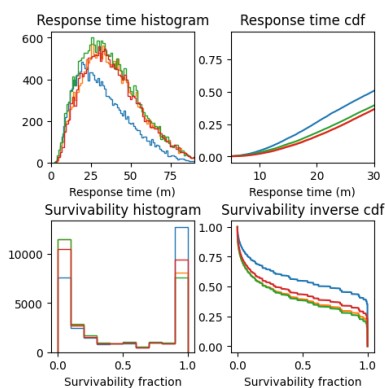

Figure 4: *Shows results for the different policies evaluated (**ex 2**): RL (Blue), Haversine (Green), Euclidean (Orange) and Random (Red).   Shows both histogram and CDF for response time. Inverse CDF is shown for survivability*

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

## A  APPENDIX

