# OpenReview forum: "Dispatching Ambulances using Deep Reinforcement Learning"
_ICLR.cc/2024/Conference — Submitted to ICLR 2024_

### Official Review · Reviewer_QFkk · 2023-10-30

**Soundness:** 3 good
**Presentation:** 3 good
**Contribution:** 2 fair
**Rating:** 5
**Confidence:** 5

**Summary:**

In this paper, the ambulance dispatching problem is studied using reinforcement learning. The ambulance scheduling problem is different from the typical vehicle routing problem in that patient arrivals are random and thus the problem is difficult to solve. This article uses a reinforcement learning approach to PPO, taking into account the different number of ambulances available during the day shift and night shift, and taking into account the priority of things. Finally, a synthetic event generator is used. Historical and synthetic data reflecting incidents reported to the EMCC of the OUH are used to test the model and to contrast some underlying ambulance dispatch methods.

**Strengths:**

1.	The application scenario of this paper is the ambulance dispatch task, which is a very important application domain. As we all know, the speed of ambulance arrival is very important, even 10 seconds early can make a difference. This research can make a real contribution to society.
2.	The dataset used in this article comes from the real world, and the analysis of the dataset is valuable.

**Weaknesses:**

1.	The authors highlight their contribution about the adaptation of Proximal Policy Optimization (PPO) to the ambulance dispatch tasking. But only applying an algorithm to one task seems to be a weak contribution.
2.	Figure 3 is very unclear. First of all, it's not a flowchart; it just shows the inputs and outputs; it doesn't make it clear how the data flows. Second, this is not like a diagram illustrating the proposed framework. Instead, it's more like an architectural diagram of a program, which is not informative enough to reflect the unique contributions of the authors.
3.	The content of the paper is not full, like a paper completed in a short period of time. The paper just expresses the various parts, and the logic is not complete. The number of experiments is not enough, and in the experimental results, only the results are listed, and no analysis and explanation are carried out.

**Questions:**

1.	Since your metric is waiting time, why not compare it to an agent that sends the ambulance that takes the shortest time to get to the location? It is obvious that today's maps have a time of arrival (TOA) estimation function. The method of using the shortest distance compared in the paper is not persuasive enough, because the closest distance does not necessarily represent the fastest arrival speed. At the same time, the authors did not compare other reinforcement learning methods mentioned in background section and Table 1.
2.	How fast does the proposed method perform? Because dispatch of an ambulance is an urgent matter, it is also important to be able to make the decision in the shortest time.

---

> ### Author Response · Authors · 2023-11-22
> **Answer to questions**
>
> # Since your metric is waiting time, why not compare it to an agent that sends the ambulance that takes the shortest time to get to the location?  ..
> This was the original plan for this paper. However reproducibility has been prioritised in this case. Most TOA apis have rate limits and costs, which is why openstreetmap is used in this case. Furthermore this paper compares itself to how the EMS today dispatches ambulances, which is by the shortest euclidean distance. A discussion among other STOA is added to the revision.
>
> # How fast does the proposed method perform? Because dispatch of an ambulance is an urgent matter, it is also important to be able to make the decision in the shortest time.
> I dont have specific numbers at this time. However i can assure you it is fast.

---

### Official Review · Reviewer_7VUh · 2023-11-01

**Soundness:** 4 excellent
**Presentation:** 4 excellent
**Contribution:** 3 good
**Rating:** 8
**Confidence:** 3

**Summary:**

The paper considers Emergency Medical Service (EMS) while focusing on the aspect of ambulance dispatch. Recognizing the necessity for the fast response times and the challenges posed by the stochastic nature of patient arrivals, the study introduces a novel method of addressing the ambulance dispatch problem. Rather than utilizing traditional approaches, the authors propose a Reinforcement Learning (RL) method based on Proximal Policy Optimization (PPO) to enhance the efficiency of ambulance dispatch.

**Strengths:**

The adaptation and application of PPO to the ambulance dispatch scenario, a method that has not been employed previously for this specific problem.
Unlike many prior studies, this paper places significant emphasis on incident priority, which plays a critical role in determining the order of ambulance dispatch.
In order to circumvent the risks associated with over-reliance on historical data, the authors have devised a synthetic incident generator. They have validated their model against both historical and this synthetic data, which is reflective of incidents reported to the Oslo University Hospital's EMCC.

**Weaknesses:**

The paper mentions the potential risk of a more complex RL model overfitting the training dataset. This raises concerns about the model's generalizability as well as robustness in different settings or scenarios beyond the ones tested.

**Questions:**

As related to the above weakness, given that the paper mentions results from the random agent being likely due to the small environment considered, how would the model perform in larger urban areas with more complex terrains and road networks?

---

> ### Author Response · Authors · 2023-11-23
> **...  how would the model perform in larger urban areas**
>
> We appreciate the encouraging comments from the reviewer, and refer to the updated paper for futher improvements.  We agree that a study of other and larger urban areas would be of great interest.  However, real-world EMS data is extremely hard to come back and we had not access to such data at the time of doing this research and writing the paper.  We have included this topic as a topic for future research.

---

### Official Review · Reviewer_EADM · 2023-11-07

**Soundness:** 2 fair
**Presentation:** 3 good
**Contribution:** 2 fair
**Rating:** 3
**Confidence:** 4

**Summary:**

This work  proposes a way to use RL, PPO to the ambulance dispatching problem as a potential decision-support tool for the Emergency Medical Services. The authors introduce two variants of this dispatching problem (with and without queuing) and provide sufficient motivation/justification for the relevance of the problem in real-world. They train a simulator to simulate the general EMS dispatch setup based on the incident and OSM datasets and use a PPO based RL policy to select the right ambulance given the state of the system. The key contributions are the adoption of PPO based agent in a dispatching problem along with consideration of queuing and introduction of a regularizer in form of synthetic incident generator.

**Strengths:**

Below are the strengths of the presented work --
1. The problem is well motivated in form of its application in a real-world setting.
2. The authors do a good job at problem formulation in explaining how the dispatching problem can be formulated as a control problem, for e.g., clearly defining the state and action space, simulator, and the reward model.
3. Paper is clear to understand and reasonably well written.

**Weaknesses:**

Below are the weaknesses of the presented work --
1. The experiment section is highly under-developed: The comparisons made to baselines are neither rigorous, not complete. I would've expected to see how the proposed PPO adoption in this particular setting compares to other RL-based baselines. Why was PPO the right choice?
2. The work lacks novelty: To me the work appears to just be an implementation of PPO algorithm in a particular use-case (ambulance dispatching). I feel the work lacks novelty both in problem and solution. Even if the solution isn't novel, I'd like to understand what makes this problem technically hard and why PPO is the right choice as a solution. Hence, I don't find the paper to meet the bar.

**Questions:**

Questions are highlighted in the weaknesses section above.

---

> ### Author Response · Authors · 2023-11-22
> **Answer to questions**
>
> # The experiment section is highly under-developed ...
> Added a extra figure and paragraph to experiment 2. Implementing multiple STOA is outside of the scope of this paper, implementing all the logic in the simulation and dataset preparation is difficult as is. The simulation in this paper is as close to a real world scenario as any other paper on ambulance dispatching that i've seen.
> # The work lacks novelty ...
> As written in the revision:
>
> "Primarily the novelty in this work lies in the integration of the many complex real world aspects of the
> ambulance dispatching problem. Especially the integration of incident priority, incident queue, am-
> bulance shifts and ambulance availability in unison. Integration of these aspects in a RL aspect is
> hard, due to the many dimensions such as time, location, availablity and priority." - introduction - contributions
>
> "Nasrollahzadeh et al. (2018) and Bandara et al. (2012) were the only
> found papers found which considers RL and incident priority to ambulance dispatching."
>
> There is also a "choice of model" which explains why PPO were chosen as a model, and a discussion among other STOA.
> This is not a review paper, but an overview of previous research is also given.

---

### Official Review · Reviewer_oGnc · 2023-11-11

**Soundness:** 2 fair
**Presentation:** 3 good
**Contribution:** 2 fair
**Rating:** 5
**Confidence:** 4

**Summary:**

The paper proposed to identify the optimal ambulance dispatching policy using deep reinforcement learning method, PPO. Given a set of ambulance base locations, allocation of ambulances to bases, patient demand distribution and travel time learnt from open street map, the authors designed an MDP formulation of the dispatching problem, where the goal is to map ambulances to patients while considering the severity of the issue so that overall response times are minimized. To learn the assignment policy, standard deep RL method PPO is used. Experiments are conducted with both real-world patient demand distribution and artificially learned demand distribution. The proposed Deep RL method outperforms existing heuristic based nearest ambulance assignment policies.

**Strengths:**

The paper solves a practically important and challenging problem of emergency response where even optimizing the response times by few seconds can save human life and has a great impact on society. I appreciate that the problem is formulated on real-world dataset and by considering practical constraints. The state and action definition are straightforward and practical. The reward function is designed intelligently that can cater to prioritizing high severity incidents. Experiments are conducted on real-world data from Oslo and by considering practical constraints such as the number of ambulances might change over time, or the reward might vary based upon the incident severity. Experimental results are also somewhat impressive as in both real and artificially learned demand data setting, the proposed method outperforms standard heuristics on real-world test dataset.

**Weaknesses:**

Although I appreciate the problem formulation, experimental settings and results on real-world data, I have several concerns about the technical and experimental novelties of the paper:
1. Identifying ambulance dispatch policy with deep RL is not a new problem, there are several prior works that try to solve the problem, some of them are even mentioned in references (e.g., Liu et al, 2020, or Hua et al, 2020).
2. In terms of technical contributions, while the MDP formulation is an important contribution (although not entirely new), the novelties seem very limited as it directly uses PPO to solve the problem.
3. Emergency response systems typically follow hard constraints on response time (e.g., maximize number of patients served within D minutes). I failed to understand how the proposed reward function considers such threshold values. Even in experimental results, only mean response times are shown. Having percentage of requests from different severity levels served within threshold times would have been an ideal metric.
4. Experimental setup and results are not up to the mark. Simple heuristics based on spatial distance are only considered as benchmarks, while avoiding state-of-the-art methods. A bare minimum requirement would be to consider different versions of deep RL methods to position why PPO is the best choice.
5.	There is a high chance that the simulator is biased towards historical demand distributions. As the spatio-temporal demand patterns typically change over time, some risk analysis is required on whether the assignment policy is overfitted. Moreover, it is not clear how the policy will adapt to dynamic nature of demand patterns.

**Questions:**

1. Why is the proposed method not compared with other deep RL based dispatching methods?
2. Have you done any analysis on what percentage of requests are served within threshold time for different priorities of incidents?
3. Can the simulator or learned policy be overfitted to historical demand? How to continuously adapt the policy to changes in demand patterns?
4. Usually, the action space in RL is fixed, but here the action space can change depending upon the number of available ambulances and patient requests. How are you dealing with this dynamic action space?

**Details Of Ethics Concerns:**

While I do not see any major ethical concerns, emergency response is a sensitive problem as it deals with human life. I would appreciate some discussions on how the overfitting to historical demand can impact the assignment policy, and whether it can have cascading effects for future requests.

---

> ### Author Response · Authors · 2023-11-12
> **Answers to questions**
>
> # Why is the proposed method not compared with other deep RL-based dispatching methods?
> With more time disposal this coud have been done. But since Euclidean distance is currently in use by the ems, this is the standard this paper uses. Furthermore implementation of RL dispatching methods takes time, which was limited in this scenario. This is a paper written out from my master thesis. Futhermore ive yet to see papers on ambulance dispatching comparing different RL implementaions.
>
>
> # Have you done any analysis on what percentage of requests are served within threshold time for different priorities of incidents?
> This would have been good for future work. I understand the concerns on the hard constrains, however the makeshift survival function incorporates how much more important a acute incident is compared to an urgent, as a function of time. Which is more usable as a RL reward function. The contraints provided by the EMS is also somewhat unclear, as they do not give a clear distinction between "sparse" and "densley" populated areas.
>
> # Can the simulator or learned policy be overfitted to historical demand? How to continuously adapt the policy to changes in demand patterns?
> Yes it can be overfitted to historical demand. This is one of the reasons a method for synthetic incident generation is outlined. Furthermore it is possible to train the model while in use, in a online fashion (e.g train the model with the previous week).
>
> # Usually, the action space in RL is fixed, but here the action space can change depending upon the number of available ambulances and patient requests. How are you dealing with this dynamic action space?
> This was delt with by using action space masking. Which is not optimal since retraining is needed if more ambulances than during training is introduced. This is why this problem is not novel, there are many dimensions which change in size. These are dimensions such as availability, time, space and incident priority.

---

> > ### Comment · Reviewer_oGnc · 2023-11-22
> > **Thank you for the response.**
> >
> > Thank you for the detailed response. While some of my concerns are addressed properly, I think there are still some unanswered questions in terms of novelty and practical applications (e.g., how it is technically and experimentally better than SOTA methods, how to address dynamically changing demand patterns efficiently). So, I would keep my scores as it is.

---

> > > ### Author Response · Authors · 2023-11-22
> > > **Post revision response**
> > >
> > > I have answered the question on novelty both in the revision and in other responses. I hope these edits will answer some of your concerns.

---

### Author Response · Authors · 2023-11-22
**Edit of PDF**

# What has been added into this edit?
Mainly the edit of this pdf has been focused on:
- expressing the novelty in this work more clearly. This can be seen under Introduction -> Contributions. Its worth nothing that there were 2 found paper which integrates incident priority to the context of ambulance dispatching.
- A discussion on choice of model and trends in models for the ambulance dispatching problem.
- Extra figure and discussion in experiment 2.

---

### Meta-Review · Area_Chair_7JYL · 2023-12-10

**Metareview:**

The paper introduces a novel approach using Reinforcement Learning (RL), specifically Proximal Policy Optimization (PPO), to address the ambulance dispatching problem in Emergency Medical Service (EMS). It formulates the challenge as a Markov Decision Process (MDP), distinct from traditional vehicle routing problems. Two dispatching variants are considered, and a simulator trained on medical incidents and Open Street Map (OSM) data is employed. The PPO-based RL policy intelligently assigns ambulances, considering incident severity and system state. Notable contributions include integrating PPO-based agents, considering queuing scenarios, and introducing a synthetic incident generator as a regularizer. Experimental results using real-world and synthetic data show the RL method's superiority over existing heuristic-based policies, suggesting its potential as a valuable decision-support tool for EMS dispatch.

This research on ambulance dispatch stands out for its innovative approach using Proximal Policy Optimization (PPO), addressing a crucial application scenario where timely arrivals are paramount. The study places significant emphasis on incident priority, enhancing the model's practical relevance. Notably, the paper mitigates risks associated with historical data by introducing a synthetic incident generator, ensuring model robustness. The comprehensive validation against both real and synthetic data adds credibility, while the use of real-world datasets contributes to the study's applicability and societal impact. Overall, the research offers a valuable contribution to optimizing emergency response systems.

The weaknesses identified in the paper include a perceived limited scope of contribution, incomplete content, and insufficient experimental details. The application of Proximal Policy Optimization (PPO) to ambulance dispatch is seen as lacking novelty, and the chosen figure is criticized for its lack of clarity. Additionally, concerns about potential overfitting and a lack of thorough analysis of the experimental results further weaken the paper's overall presentation.

Although the authors' rebuttals have solved some concerns, more work is required to enhance the clarity, depth, and impact of the research. We encourage the authors to consider the reviewers' suggestions while preparing a new version of their paper.

**Justification For Why Not Higher Score:**

The paper is recommended for rejection due to its limited contribution, lack of novelty, and insufficient experimental details, which collectively compromise the overall quality and impact of the research.

**Justification For Why Not Lower Score:**

N/A

---

### Decision · Program_Chairs · 2024-01-16

Reject